# ON SYMMETRY AND INITIALIZATION FOR NEURAL NETWORKS

## ABSTRACT

This work provides an additional step in the theoretical understanding of neural networks. We consider neural networks with one hidden layer and show that when learning symmetric functions, one can choose initial conditions so that standard SGD training efficiently produces generalization guarantees. We empirically verify this and show that this does not hold when the initial conditions are chosen at random. The proof of convergence investigates the interaction between the two layers of the network. Our results highlight the importance of using symmetry in the design of neural networks.

## 1 INTRODUCTION

Building a theory that can help to understand neural networks and guide their construction is one of the current challenges of machine learning. Here we wish to shed some light on the role symmetry plays in the construction of neural networks. It is well-known that symmetry can be used to enhance the performance of neural networks. For example, convolutional neural networks (CNNs) (see Lecun et al. (1998)) use the translational symmetry of images to classify images better than fully connected neural networks. Our focus is on the role of symmetry in the initialization stage. We show that symmetry-based initialization can be the difference between failure and success.

On a high-level, the study of neural networks can be partitioned to three different aspects.

***Expressiveness*** Given an architecture, what are the functions it can approximate well?

***Training*** Given a network with a "proper" architecture, can the network fit the training data and in a reasonable time?

***Generalization*** Given that the training seemed successful, will the true error be small as well?

We study these aspects for the first "non trivial" case of neural networks, networks with one hidden layer. We are mostly interested in the initialization phase. If we take a network with the appropriate architecture, we can always initialize it to the desired function. A standard method (that induces a non trivial learning problem) is using random weights to initialize the network. A different reasonable choice is to require the initialization to be useful for an entire class of functions. We follow the latter option.

Our focus is on the role of symmetry. We consider the following class of symmetric functions

$$\mathbb{S} = \mathbb{S}_n = \Big\{ \sum_{i=0}^{n} a_i \cdot \mathbb{1}_{|x|=i} : a_1, \ldots, a_n \in \{\pm 1\} \Big\},$$

where $x \in \{0,1\}^n$ and $|x| = \sum_i x_i$. The functions in this class are invariant under arbitrary permutations of the input's coordinates. The parity function $\pi(x) = (-1)^{|x|}$ and the majority function are well-known examples of symmetric functions.

***Expressiveness*** for this class was explored by Minsky and Papert (1988). They showed that the parity function cannot be represented using a network with limited "connectivity". Contrastingly, if we use a fully connected network with one hidden layer and a common activation function (like sign, sigmoid, or ReLU) only $O(n)$ neurons are needed. We provide such explicit representations for all functions in $\mathbb{S}$; see Lemmas 1 and 2.

We also provide useful information on both the *training* phase and *generalization* capabilities of the neural network. We show that, with proper initialization, the training process (using standard SGD) efficiently converges to zero empirical error, and that consequently the network has small true error as well.

**Theorem 1.** *There exists a constant $c > 1$ so that the following holds. There exists a network with one hidden layer, $cn$ neurons with* sigmoid *or* ReLU *activations, and an initialization such that for all distributions $\mathscr{D}$ over $X = \{0,1\}^n$ and all functions $f \in \mathbb{S}$ with sample size $m \geq c(n + \log(1/\delta))/\varepsilon$, after performing $poly(n)$ SGD updates with a fixed step size $h = 1/poly(n)$ it holds that*

$$\underset{x^m \sim \mathscr{D}^m}{P}\left(\left\{S: \underset{x \sim \mathscr{D}}{\Pr}\left(N_S(x) \neq f(x)\right) > \varepsilon\right\}\right) < \delta$$

*where $S = \{(x_1, f(x_1)), ..., (x_m, f(x_m))\}$ and $N_S(x)$ is the network after training over $S$.*

The number of parameters in the network described in Theorem 1 is $\Omega(n^2)$. So in general one could expect overfitting when the sample size is as small as $O(n)$. Nevertheless, the theorem provides generalization guarantees, even for such a small sample size.

The initialization phase plays an important role in proving Theorem 1. To emphasize this, we report an empirical phenomenon (this is "folklore"). We show that a network cannot learn parity from a random initialization (see Section 5.3). On one hand, if the network size is big, we can bring the empirical error to zero (as suggested in Soudry and Carmon (2016)), but the true error is close to $1/2$. On the other hand, if its size is too small, the network is not even able to achieve small empirical error (see Figure 5). We observe a similar phenomenon also for a random symmetric function. An open question remains: why is it true that a sample of size polynomial in $n$ does not suffice to learn parity (with random initialization)?

A similar phenomenon was theoretically explained by Shamir (2016) and Song et al. (2017). The parity function belongs to the class of all parities

$$\mathbb{P} = \mathbb{P}_n = \{\pi_s(x) = (-1)^{s \cdot x} : s \in X\}$$

where $\cdot$ is the standard inner product. This class is efficiently PAC-learnable with $O(n)$ samples using Gaussian elimination. A continuous version of $\mathbb{P}$ was studied by Shamir (2016) and Song et al. (2017). To study the training phase, they used a generalized notion of *statistical queries* (SQ); see Kearns (1998). In this framework, they show that most functions in the class $\mathbb{P}$ cannot be efficiently learned (roughly stated, learning the class requires an exponential amount of resources). This framework, however, does not seem to capture actual training of neural networks using SGD. For example, it is not clear if one SGD update corresponds to a single query in this model. In addition, typically one receives a dataset and performs the training by going over it many times, whereas the query model estimates the gradient using a fresh batch of samples in each iteration. The query model also assumes the noise to be adversarial, an assumption that does not necessarily hold in reality. Finally, the SQ-based lower bound holds for every initialization (in particular, for the initialization we use here), so it does not capture the efficient training process Theorem 1 describes.

Theorem 1 shows, however, that with symmetry-based initialization, parity can be efficiently learned. So, in a nutshell, parity can not be learned as part of $\mathbb{P}$, but it can be learned as part of $\mathbb{S}$. One could wonder why the hardness proof for $\mathbb{P}$ cannot be applied for $\mathbb{S}$ as both classes consist of many input sensitive functions. The answer lies in the fact that $\mathbb{P}$ has a far bigger statistical dimension than $\mathbb{S}$ (all functions in $\mathbb{P}$ are orthogonal to each other, unlike $\mathbb{S}$).

The proof of the theorem utilizes the different behavior of the two layers in the network. SGD is performed using a step size $h$ that is polynomially small in $n$. The analysis shows that in a polynomial number of steps that is *independent* of the choice of $h$ the following two properties hold: (i) the output neuron reaches a "good" state and (ii) the hidden layer does not change in a "meaningful" way. These two properties hold when $h$ is small enough. In Section 5.2, we experiment with large values of $h$. We see that, although the training error is zero, the true error becomes large.

Here is a high level description of the proof. The $\ell$ neurons in the hidden layer define an "embedding" of the inputs space $X = \{0,1\}^n$ into $\mathbb{R}^\ell$ (a.k.a. the feature map). This embedding changes in time according to the training examples and process. The proof shows that if at any point in time this embedding has good enough margin, then training with standard SGD quickly converges. This is explained in more detail in Section 3. It remains an interesting open problem to understand this phenomenon in greater generality, using a cleaner and more abstract language.

## 1.1 BACKGROUND

To better understand the context of our research, we survey previous related works.

The expressiveness and limitations of neural networks were studied in several works such as Rahimi and Recht (2008); Telgarsky (2016); Eldan and Shamir (2016) and Arora et al. (2016). Constructions of small ReLU networks for the parity function appeared in several previous works, such as Wilamowski et al. (2003), Arslanov et al. (2016), Arslanov et al. (2002) and Masato Iyoda et al. (2003). Constant depth circuits for the parity function were also studied in the context of computational complexity theory, see for example Furst et al. (1981), Ajtai (1983) and Håstad (1987).

The training phase of neural networks was also studied in many works. Here we list several works that seem most related to ours. Daniely (2017) analyzed SGD for general neural network architecture and showed that the training error can be nullified, e.g., for the class of bounded degree polynomials (see also Andoni et al. (2014)). Jacot et al. (2018) studied neural tangent kernels (NTK), an infinite width analogue of neural networks. Du et al. (2018) showed that randomly initialized shallow ReLU networks nullify the training error, as long as the number of samples is smaller than the number of neurons in the hidden layer. Their analysis only deals with optimization over the first layer (so that the weights of the output neuron are fixed). Chizat and Bach (2018) provided another analysis of the latter two works. Allen-Zhu et al. (2018b) showed that over-parametrized neural networks can achieve zero training error, as as long as the data points are not too close to one another and the weights of the output neuron are fixed. Zou et al. (2018) provided guarantees for zero training error, assuming the two classes are separated by a positive margin.

Convergence and generalization guarantees for neural networks were studied in the following works. Brutzkus et al. (2017) studied linearly separable data. Li and Liang (2018) studied well separated distributions. Allen-Zhu et al. (2018a) gave generalization guarantees in expectation for SGD. Arora et al. (2019) gave data-dependent generalization bounds for GD. All these works optimized only over the hidden layer (the output layer is fixed after initialization).

Margins play an important role in learning, and we also use it in our proof. Sokolic et al. (2016), Sokolic et al. (2017), Bartlett et al. (2017) and Sun et al. (2015) gave generalization bounds for neural networks that are based on their margin when the training ends. From a practical perspective, Elsayed et al. (2018), Romero and Alquezar (2002) and Liu et al. (2016) suggested different training algorithms that optimize the margin.

As discussed above, it seems difficult for neural networks to learn parities. Song et al. (2017) and Shamir (2016) demonstrated this using the language statistical queries (SQ). This is a valuable language, but it misses some central aspects of training neural networks. SQ seems to be closely related to GD, but does not seem to capture SGD. SQ also shows that many of the parities functions $\otimes_{i \in S} x_i$ are difficult to learn, but it does not imply that *the* parity function $\otimes_{i \in [n]} x_i$ is difficult to learn. Abbe and Sandon (2018) demonstrated a similar phenomenon in a setting that is closer to the "real life" mechanics of neural networks.

We suggest that taking the symmetries of the learning problem into account can make the difference between failure and success. Several works suggested different neural architectures that take symmetries into account; see Zaheer et al. (2017), Gens and Domingos (2014), and Cohen and Welling (2016).

## 2 REPRESENTATIONS

Here we describe efficient representations for symmetric functions by networks with one hidden layer. These representations are also useful later on, when we study the training process. We study two different activation functions, sigmoid and ReLU (similar statement can be proved for other activations, like arctan). Each activation function requires its own representation, as in the two lemmas below.

### 2.1 SIGMOID

We start with the activation $\sigma(\xi) = \frac{1}{1+\exp(-\xi)}$, since it helps to understand the construction for the ReLU activation. The building blocks of the symmetric functions are indicators of $|x| = i$ for

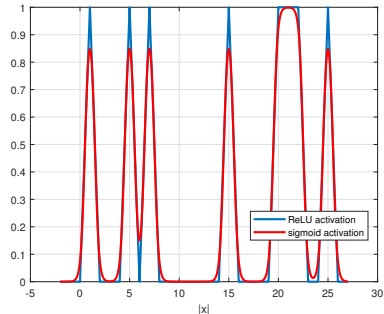

Figure 1: Approximations of the symmetric function $f_A = \text{sign}(\sum_{i \in A} \mathbb{1}_{|x|=i} - 0.5)$ by sigmoid and ReLU activations for $A = \{1, 5, 7, 15, 20, 21, 22, 25\}$.

$i \in \{0, 1, \ldots, n\}$. An indicator function is essentially the difference between two sigmoid functions:

$$\text{sign}(\mathbb{1}_{|x|=i} - 0.5) = \text{sign}(\Delta_i - 0.5),$$

where $\Delta_i(x) = \sigma(5(|x| - i + 0.5)) - \sigma(5(|x| - i - 0.5))$.

**Lemma 1.** *The symmetric function $f_A = \text{sign}(\sum_{i \in A} \mathbb{1}_{|x|=i} - 0.5)$ satisfies $f_A(x) = \text{sign}(-0.5 + \sum_{i \in A} \Delta_i(x))$, where $A \subset [n]$.*

A network with one hidden layer of $n + 2$ neurons with sigmoid activations and one bias neuron is sufficient to represent any function in $\mathbb{S}$. The coefficients of the sigmoid gates are $0, \pm 1$ in this representation. The proofs of this lemma and the subsequent lemmas appear in the appendix.

## 2.2 RELU

A sigmoid function can be represented using $\text{ReLU}(\xi) = \max\{0, \xi\}$ as the difference between two ReLUs

$$\sigma(5(x + 0.5)) \approx \text{ReLU}(x + 1) - \text{ReLU}(x)$$

Hence, an indicator function can be represented using $\text{sign}(\mathbb{1}_{|x|=i} - 0.5) = \text{sign}(\Gamma_i - 0.5)$ where

$$\Gamma_i(x) = \text{ReLU}(|x| - i + 1) - 2\text{ReLU}(|x| - i) + \text{ReLU}(|x| - i - 1).$$

**Lemma 2.** *The symmetric function $f_A = \text{sign}(\sum_{i \in A} \mathbb{1}_{|x|=i} - 0.5)$ satisfies $f_A(x) = \text{sign}(-0.5 + \sum_{i \in A} \Gamma_i(x))$, where $A \subset [n]$.*

The lemma shows that a network with one hidden layer of $n + 3$ ReLU neurons and one bias neuron is sufficient to represent any function in $\mathbb{S}$. The coefficients of the ReLU gates are $0, \pm 1, \pm 2$ in this representation.

## 3 TRAINING AND GENERALIZATION

The goal of this section is to describe a small network with one hidden layer that (when initialized properly) efficiently learns symmetric functions using a small number of examples (the training is done via SGD).

### 3.1 SPECIFICATIONS

Here we specify the architecture, initialization and loss function that is implicit in our main result (Theorem 1).

To guarantee convergence of SGD, we need to start with "good" initial conditions. The initialization we pick depends on the activation function it uses, and is chosen with resemblance to Lemma 2 for

ReLU. On a high level, this indicates that understanding the class of functions we wish to study in term of "representation" can be helpful when choosing the architecture of a neural network in a learning context.

The network we consider has one hidden layer. We denote by $w_{ij}$ the weight between coordinate $j$ of the input and neuron $i$ in the hidden layer. We denote $W$ this matrix of weights. We denote by $b_i$ the bias of neuron $i$ of the hidden layer. We denote $B$ this vector of weights. We denote by $m_i$ is the weight from neuron $i$ in the hidden layer to the output neuron. We denote $M$ this vector of weights. We denote by $b$ the bias of the output neuron.

Initialize the network as follows: The dimensions of $W$ are $(n+3) \times n$. For all $1 \leq i \leq (n+3)$ and $1 \leq j \leq n$, we set $w_{ij} = 1$ and $b_i = -i + 2$. We set $M = 0$ and $b = 0$.

To run SGD, we need to choose a loss function. We use the *hinge loss*,

$$L(x,f) = \max\{0, -f(x)(v_x \cdot M + b) + \beta\},$$

where $v_x = \mathrm{ReLU}(Wx + B)$ is the output of the hidden layer on input $x$ and $\beta > 0$ is a parameter of confidence.

## 3.2 MARGINS

A key property in the analysis is the 'margin' of the hidden layer with respect to the function being learned.

A map $Y : V \to \{\pm 1\}$ over a finite set $V \subset \mathbb{R}^d$ is linearly[1] separable if there exists $w \in \mathbb{R}^d$ such that $\mathrm{sign}(w \cdot v) = Y(v)$ for all $v \in V$. When the Euclidean norm of $w$ is $\|w\| = 1$, the number $\mathrm{marg}(w,Y) = \min_{v \in V} Y(v)w \cdot v$ is the margin of $w$ with respect to $Y$. The number $\mathrm{marg}(Y) = \sup_{w \in \mathbb{R}^d : \|w\|=1} \mathrm{marg}(w,Y)$ is the margin of $Y$.

We are interested in the following set $V$ in $\mathbb{R}^d$. Recall that $W$ is the weight matrix between the input layer and the hidden layer, and that $B$ is the relevant bias vector. Given $W, B$, we are interested in the set $V = \{v_x : x \in X\}$, where $v_x = \mathrm{ReLU}(Wx + B)$. In words, we think of the neurons in the hidden layer as defining an "embedding" of $X$ in Euclidean space. A similar construction works for other activation functions. We say that $Y : V \to \{\pm 1\}$ agrees with $f \in \mathbb{S}$ if for all $x \in X$ it holds that $Y(v_x) = f(x)$.

The following lemma bounds from below the margin of the initial $V$.

**Lemma 3.** *If $Y$ is a partition that agrees with some function in $\mathbb{S}$ for the initialization described above then $\mathrm{marg}(Y) \geq \Omega(1/\sqrt{n})$.*

*Proof.* By Lemmas 1 and 2, we see that any function in $\mathbb{S}$ can be represented with a vector of weights $M, b \in [-1,1]^{\Theta(n)}$ of the output neuron together with a bias . These $M, b$ induce a partition $Y$ of $V$. Namely, $Y(v_x)M \cdot v_x + b > 0.25$ for all $x \in X$. Since $\|(M,b)\| = O(\sqrt{n})$ we have our desired result. $\qquad \square$

## 3.3 FREEZING THE HIDDEN LAYER

Before analyzing the full behavior of SGD, we make an observation: if the weights of the hidden layer are fixed with the initialization described above, then Theorem 1 holds for SGD with batch size 1. This observation, unfortunately, does not suffice to prove Theorem 1. In the setting we consider, the training of the neural network uses SGD without fixing any weights. This more general case is handled in the next section. The rest of this subsection is devoted for explaining this observation.

Novikoff (1962) showed that that the perceptron algorithm Rosenblatt (1958) makes a small number of mistakes for linearly separable data with large margin. For a comprehensive survey of the perceptron algorithm and its variants, see Moran et al. (2018).

Running SGD with the hinge loss induces the same update rule as in a modified perceptron algorithm, Algorithm 1.

---

[1] A standard "lifting" that adds a coordinate with 1 to every vector allows to translate the affine case to the linear case.

---

**Algorithm 1** The modified perceptron algorithm

---

    **Initialize:** $w^{(0)} = \vec{0}$, $t = 0$, $\beta > 0$ and $h > 0$
    **while** $\exists v \in V$ with $Y(v)w^{(t)} \cdot v \leq \beta$ **do**
        $w^{(t+1)} = w^{(t)} + Y(v)vh$
        $t = t + 1$
    **end while**
    **return** $w^{(t)}$

---

Novikoff's proof can be generalized to any $\beta > 0$ and batches of any size to yield the following theorem; see Collobert and Bengio (2004); Krauth and Mezard (1987) and appendix A.

**Theorem 2.** *For $Y : V \to \{\pm 1\}$ with margin $\gamma > 0$ and step size $h > 0$, the modified perceptron algorithm performs at most $\frac{2\beta h + (Rh)^2}{(\gamma h)^2}$ updates and achieves a margin of at least $\frac{\gamma \beta h}{2\beta h + (Rh)^2}$, where $R = \max_{v \in V} \|v\|$.*

So, when the weights of the hidden layer are fixed, Lemma 3 implies that the number of SGD steps is at most polynomial in $n$.

### 3.4 STABILITY

When we run SGD on the entire network, the layers interact. For a ReLU network at time $t$, the update rule for $W$ is as follows. If the network classifies the input $x$ correctly with confidence more than $\beta$, no change is made. Otherwise, we change the weights in $M$ by $\Delta M = yv_x h$, where $y$ is the true label and $h$ is the step size. If also neuron $i$ of the hidden *fired* on $x$, we update its incoming weights by $\Delta W_{i,:} = ym_i xh$. These update rules define the following dynamical system: $(a)$

$$W^{(t+1)} = W^{(t)} + y\left(\left(M^{(t)}\right)\right) \tag{1}$$

$$W^{(t+1)} = W^{(t)} + y\left(\left(M^{(t)}\right)^T \circ H\left(W^{(t)}x + B^{(t)}\right)\right)x^T h \tag{2}$$

$$B^{(t+1)} = B^{(t)} + y\left(\left(M^{(t)}\right)^T \circ H\left(W^{(t)}x + B^{(t)}\right)\right)h \tag{3}$$

$$M^{(t+1)} = M^{(t)} + y\,\text{ReLU}\left(W^{(t)}x + B^{(t)}\right)h \tag{4}$$

$$b^{(t+1)} = b^{(t)} + yh, \tag{5}$$

where $H$ is the Heaviside step function and $\circ$ is the Hadamard pointwise product.

A key observation in the proof is that the weights of the last layer ((4) and (5)) are updated exactly as the modified perceptron algorithm. Another key statement in the proof is that if the network has reached a good representation of the input (i.e., the hidden layer has a large margin), then the interaction between the layers during the continued training does not impair this representation. This is summarized in the following lemma (we are not aware of a similar statement in the literature).

**Lemma 4.** *Let $M = 0$, $b = 0$, and $V = \{\text{ReLU}(Wx + B) : x \in X\}$ be a linearly separable embedding of $X$ and with margin $\gamma > 0$ by the hidden layer of a neural network of depth two with $\text{ReLU}$ activation and weights given by $W, B$. Let $R_X = \max_{x \in X} \|x\|$, let $R = \max_{v \in V} \|v\|$, and $0 < h \leq \frac{\gamma^{5/2}}{100R^2 R_X}$ be the integration step. Assuming $R_X > 1$ and $\gamma \leq 1$, and using $\beta = R^2 h$ in the loss function, after $t$ SGD iterations the following hold:*

    – *Each $v \in V$ moves a distance of at most $O(R_X^2 h^2 R t^{3/2})$.*

    – *The norm $\|M^{(t)}\|$ is at most $O(Rh\sqrt{t})$.*

    – *The training ends in at most $O(R^2/\gamma^2)$ SGD updates.*

Intuitively, this type of lemma can be useful in many other contexts. The high level idea is to identify a "good geometric structure" that the network reaches and enables efficient learning.

## 4 MAIN RESULT

*Proof of Theorem 1.* There is an unknown distribution $\mathcal{D}$ over the space $X$. We pick i.i.d. examples $S = ((x_1, y_1), ..., (x_m, y_m))$ where $m \geq c\left(\frac{n + \log(1/\delta)}{\varepsilon}\right)$ according to $\mathcal{D}$, where $y_i = f(x_i)$ for some $f \in \mathbb{S}$. Run SGD for $O(n^4)$ steps, where the step size is $h = O(1/n^6)$ and the parameter of the loss function is $\beta = R^2 h$ with $R = n^{3/2}$.

We claim that it suffices to show that at the end of the training (i) the network correctly classifies all the sample points $x_1, ..., x_m$, and (ii) for every $x \in X$ such that there exists $1 \leq i \leq m$ with $|x| = |x_i|$, the network outputs $y_i$ on $x$ as well. Here is why. The initialization of the network embeds the space $X$ into $n + 4$ dimensional space (including the bias neuron of the hidden layer). Let $V^{(0)}$ be the initial embedding $V^{(0)} = \{\text{ReLU}(W^{(0)}x + B^{(0)}) : x \in X\}$. Although $|X| = 2^n$, the size of $V^{(0)}$ is $n + 1$. The VC dimension of all the boolean functions over $V^{(0)}$ is $n + 1$. Now, $m$ samples suffice to yield $\varepsilon$ true error for an ERM when the VC dimension is $n + 1$; see e.g. Theorem 6.7 in Shalev-Shwartz and Ben-David (2014). It remains to prove (i) and (ii) above.

By Lemma 3, at the beginning of the training, the partition of $V^{(0)}$ defined by the target $f \in \mathbb{S}$ has a margin of $\gamma = \Omega(1/\sqrt{n})$. We are interested in the eventual $V^* = \{\text{ReLU}(W^*x + B^*) : x \in X\}$ embedding of $X$ as well. The modified perceptron algorithm together with Lemma 4 guarantees that after $K \leq 20R^2/\gamma^2 = O(n^4)$ updates, $(M^*, b^*)$ separates the embedded sample $V_S^* = \{\text{ReLU}(W^*x_i + B^*) : 1 \leq i \leq m\}$ with a margin of at least $0.9\gamma/3$.

It remains to prove (ii). Lemma 4 states that as long as less than $K = O(n^4)$ updates were made, the elements in $V$ moved at most $O(1/n^2)$. At the end of the training, the embedded sample $V_S$ is separated with a margin of at least $0.9\gamma/3$ with respect to the hyperplane defined by $M^*$ and $B^*$. Each $v_x^*$ for $x \in X$ moved at most $O(1/n^2) < \gamma/4$. This means that if $|x| = |x_i|$ then the network has the same output on $x$ and $x_i$. Since the network has zero empirical error, the output on this $x$ is $y_i$ as well.

A similar proof is available with sigmoid activation (with better convergence rate and larger allowed step size).

$\square$

**Remark.** *The generalization part of the above proof can be viewed as a consequence of sample compression (Littlestone and Warmuth (1986)). Although the eventual network depends on* all *examples, the proof shows that its functionality depends on at most $n + 1$ examples. Indeed, after the training, all examples with equal hamming weight have the same label.*

**Remark.** *The parameter $\beta = R^2 h$ we chose in the proof may seem odd and negligible. It is a construct in the proof that allows us to bound efficiently the distance that the elements in $V$ have moved during the training. For all practical purposes $\beta = 0$ works as well (see Figure 4).*

## 5 EXPERIMENTS

We accompany the theoretical results with some experiments. We used a network with one hidden layer of $4n + 3$ neurons, ReLU activation, and the hinge loss with $\beta = n^3 h$. In all the experiments, we used SGD with mini-batch of size one and before each epoch we randomized the sample. We observed similar behavior for larger mini-batches, other activation functions, and other loss functions. The graphs that appear in the appendix A present the training error and the true error[2] versus the epoch of the training process. In all the comparisons below, we chose a random symmetric function and a random sample from $X$.

### 5.1 THE THEORY IN PRACTICE

Figure 2 demonstrates our theoretical results and also validates the performance of our initialization. In one setting, we trained only the second layer (freezed the weights of the hidden layer) which

---

[2]We deal with high dimensional spaces, so the true error was not calculated exactly but approximated on an independent batch of samples of size $10^4$.

essentially corresponds to the perceptron algorithm. In the second setting, we trained both layers with a step size $h = n^{-6}$ (as the theory suggests). As expected, performance in both cases is similar. We remark that SGD continues to run even after minimizing the empirical error. This happens because of the parameter $\beta > 0$.

## 5.2 Overstepping the Theory

Here we experiment with two parameters in the proof, the step size $h$ and the confidence parameter $\beta$. In Figure 3, we used three different step sizes, two of which much larger than the theory suggests. We see that the training error converges much faster to zero, when the step size is larger. This fast convergence comes at the expense of the true error. For a large step size, generalization cease to hold.

Setting $\beta = n^3 h$ is a construct in the proof. Figure 4 shows that setting $\beta = 0$ does not impair the performance. The difference between theory (requires $\beta > 0$) and practice (allows $\beta = 0$) can be explained as follows. The proof bounds the worst-case movement of the hidden layer, whereas in practice an average-case argument suffices.

## 5.3 Hard to Learn Parity

Figure 5 shows that even for $n = 20$, learning parity is hard from a random initialization. When the sample size is small the training error can be nullified but the true error is large. As the sample grows, it becomes much harder for the network to nullify even the training error. With our initialization, both the training error and true error are minimized quickly. Figure 6 demonstrates the same phenomenon for a random symmetric function.

## 5.4 Corruption of Data

Our initialization also delivers satisfying results when the input data it corrupted. In figure 7, we randomly perturb (with probability $p = \frac{1}{10}$) the labels and use the same SGD to train the model. In figure 8, we randomly shift every entry of the vectors in the space $X$ by $\varepsilon$ that is uniformly distributed in $[-0.1, 0.1]^n$.

# 6 Conclusion

This work demonstrates that symmetries can play a critical role when designing a neural network. We proved that any symmetric function can be learned by a shallow neural network, with proper initialization. We demonstrated by simulations that this neural network is stable under corruption of data, and that the small step size is the proof is necessary.

We also demonstrated that the parity function or a random symmetric function cannot be learned with random initialization. How to explain this empirical phenomenon is still an open question. The works Shamir (2016) and Song et al. (2017) treated parities using the language of SQ. This language obscures the inner mechanism of the network training, so a more concrete explanation is currently missing.

We proved in a special case that the standard SGD training of a network efficiently produces low true error. The general problem that remains is proving similar results for general neural networks. A suggestion for future works is to try to identify favorable geometric states of the network that guarantee fast convergence and generalization.

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

## A    APPENDIX

PROOF OF LEMMA 1

*Proof.* For all $k \in A$ and $x \in X$ of weight $k$,

$$\sum_{i \in A} \Delta_i(k) \geq \Delta_k(x) = \sigma(5 \cdot 0.5) - \sigma(-5 \cdot 0.5) > 0.84;$$

the first inequality holds since $\Delta_i(x) \geq 0$ for all $i$ and $x$. For all $k \notin A$ and $x \in X$ of weight $k$,

$$
\begin{aligned}
\sum_{i \in A} \Delta_i(x) &= \sum_{i \in A} \sigma(5 \cdot (k-i+0.5)) - \sigma(5 \cdot (k-i-0.5)) \\
&= \sum_{k < i \in A} \sigma(5 \cdot (k-i+0.5)) + \sum_{k > i \in A} \sigma(5 \cdot (i+0.5-k)) \\
&\quad + \sum_{k < i \in A} [\sigma(5 \cdot (i+0.5-k)) - 1] + \sum_{k > i \in A} [\sigma(5 \cdot (k-i+0.5)) - 1] \\
&< \sum_{k < i \in A} \sigma(5 \cdot (k-i+0.5)) + \sum_{k > i \in A} \sigma(5 \cdot (i+0.5-k)) \\
&< 2 \sum_{i=1}^{\infty} \exp(5 \cdot (-i+0.5)) \\
&= 2 \exp(-2.5)/(1 - \exp(-5)) < 0.17;
\end{aligned}
$$

the first equality follows from the definition, the second equality follows from $\sigma(5(x+0.5)) - \sigma(5(x-0.5)) = \sigma(5(x+0.5)) + \sigma(5(-x+0.5)) - 1$ for all $x$, the first inequality neglects the negative sums, and the second inequality follows because $\exp(\xi) > \sigma(\xi)$ for all $\xi$.

$\square$

## PROOF OF LEMMA 2

*Proof.* The proof follows from two observations:

For all $i$ the support of $\Gamma_i(x) = \text{ReLU}(|x| - i + 1) - 2\text{ReLU}(|x| - i) + \text{ReLU}(|x| - i - 1)$ is $i - 1 \leq x \leq i + 1$. So for all $x$ of weight not in $A$ it holds $\sum_{i \in A} \Gamma_i(x) = 0$.

For all $i \in A$ and $x$ of weight $i$ it holds $\Gamma_i(x) = 1$.

$\square$

## PROOF OF LEMMA 4

*Proof.* We are interested in the maximal distance the embedding of an element $x \in X$ has moved from its initial embedding:

$$
\left\| v_x^{(t)} - v_x^{(0)} \right\| = \left\| \text{ReLU}(W^{(t)} x + B^{(t)}) - \text{ReLU}(W^{(0)} x + B^{(0)}) \right\| \tag{6}
$$

$$
\leq \left\| W^{(t)} - W^{(0)} \right\| R_X + \left\| B^{(t)} - B^{(0)} \right\| \tag{7}
$$

$$
\leq \sum_{k=1}^{t} \left[ R_X \left\| W^{(k)} - W^{(k-1)} \right\| + \left\| B^{(k)} - B^{(k-1)} \right\| \right]. \tag{8}
$$

To simplify equations (2)-(5) discussed above, we assume that during the optimization process the norm of the weights $W$ and $B$ grow at a maximal rate:

$$
\left\| W^{(t+1)} - W^{(t)} \right\| = \left\| y \left( \left(M^{(t)}\right)^T \circ H \left(W^{(t)} x + B^{(t)}\right)\right) x^T h \right\| \leq \left\| M^{(t)} \right\| R_X h, \tag{9}
$$

$$
\left\| B^{(t+1)} - B^{(t)} \right\| = \left\| y \left(M^{(t)}\right)^T \circ H \left(W^{(t)} x + B^{(t)}\right) h \right\| \leq \left\| M^{(t)} \right\| h; \tag{10}
$$

here the norm of a matrix is the $\ell_2$-norm.

To bound these quantities, we follow the modified perceptron proof and add another quantity to bound. That is, the maximal norm $R^{(t)}$ of the embedded space $X$ at time $t$ satisfies (by assumption $R_X > 1$)

$$
R^{(t+1)} \leq R^{(t)} + (1 + R_X^2) \left\| M^{(t)} \right\| h \leq R^{(t)} + 2 R_X^2 \left\| M^{(t)} \right\| h;
$$

we used that the spectral norm of a matrix is at most its $\ell_2$-norm.

We assume a worst-case where $R^{(t)}$ grows monotonically at a maximal rate. By the modified perceptron algorithm and choice $\beta = R^2 h$,

$$
\left\| M^{(t)} \right\| \leq \sqrt{t((R^{(t)} h)^2 + 2\beta h)} \leq \sqrt{3} R^{(t)} h \sqrt{t}.
$$

By choice of $h \leq \frac{\gamma^{5/2}}{100 R^2 R_X}$ and assuming $t \leq 20 R^2 / \gamma^2$,

$$R^{(t+1)} \leq R^{(t)} + 2\sqrt{3} R_X^2 R^{(t)} h^2 \sqrt{t} \leq R^{(t)} + \frac{2\sqrt{60}}{100^2} R^{(t)} \gamma^4 / R^3.$$

Solving the above recursive equation, it holds for all $t \leq 20 R^2 / \gamma^2$,

$$R^{(t)} \leq \left(1 + \frac{2\sqrt{60}}{100^2} \gamma^4 / R^3\right)^t R \leq \exp\left(\frac{40\sqrt{60}}{100^2} \gamma^2 / R\right) R \leq 2R.$$

Now, summing equation 8, we have

$$\left\|v_x^{(t)} - v_x^{(0)}\right\| \leq 2\sqrt{6} R_X^2 h^2 R t^{3/2},$$

since $\sum_{k=1}^{t} \sqrt{k} \leq t^{3/2}$.

So in $20 R^2 / \gamma^2$ updates, the elements embedded by the network travelled at most $\frac{2 \cdot 20^{3/2} \sqrt{6}}{100^2} \gamma^2 \leq 0.05 \gamma^2$. Hence, the samples the network received kept a margin of $0.9\gamma$ during training (by the assumption $\gamma \leq 1$). By choice of the loss function, SGD changes the output neuron as in the modified perceptron algorithm. By Theorem 2, the number of updates is at most $\frac{2R^2 + (2R)^2}{0.9\gamma^2} < 20 R^2 / \gamma^2$. So, the assumption on $t$ we made during the proof holds.

$\square$

### THE MODIFIED PERCEPTRON

*Proof of Theorem 2.* Denote by $w^*$ the optimal separating hyperplane with $\|w^*\| = 1$. It satisfies $y_i w^* \cdot x_i \geq \gamma$ for all $x_i$. By the definition,

$$w^{(t)} \cdot w^* = w^{(t-1)} \cdot w^* + y_i w^* \cdot x_i \geq \gamma h t$$

and

$$\left\|w^{(t)}\right\|^2 = \left\|w^{(t-1)}\right\|^2 + 2 y_i w^{(t-1)} x_i h + (\|x_i\| h)^2 \leq \left(2\beta h + (Rh)^2\right) t.$$

By Cauchy-Schwarz inequality, $1 \geq w^{(t)} \cdot w^* / \left\|w^{(t)}\right\|$. So the number of updates is bounded by

$$\frac{2\beta h + (Rh)^2}{(\gamma h)^2}.$$

At time $t$ the margin of any $x_i$ that does not require an update is at least

$$\frac{\beta}{\|w^{(t)}\|} \geq \frac{\beta}{\sqrt{\left(2\beta h + (Rh)^2\right) t}}.$$

The right hand side is monotonically decreasing function of $t$ so by plugging in the maximal number of updates we see that the minimal margin of the output is at least

$$\frac{\gamma \beta h}{2\beta h + (Rh)^2}.$$

$\square$

FIGURES

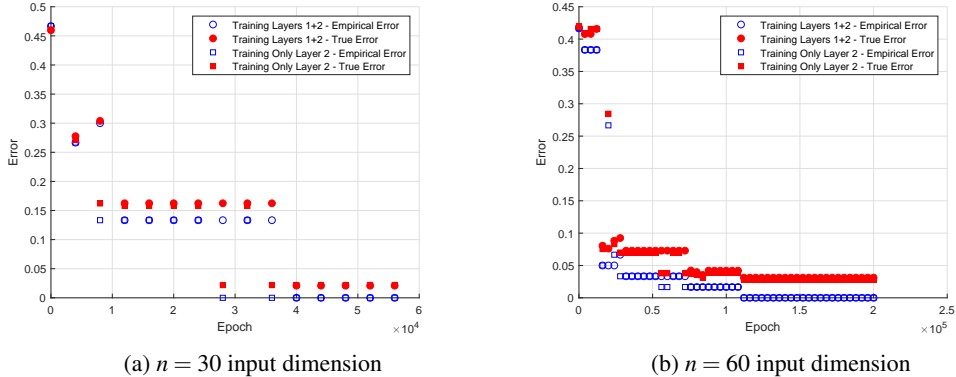

(a) $n = 30$ input dimension
(b) $n = 60$ input dimension

Figure 2: Error during training for a sample of size $n$.

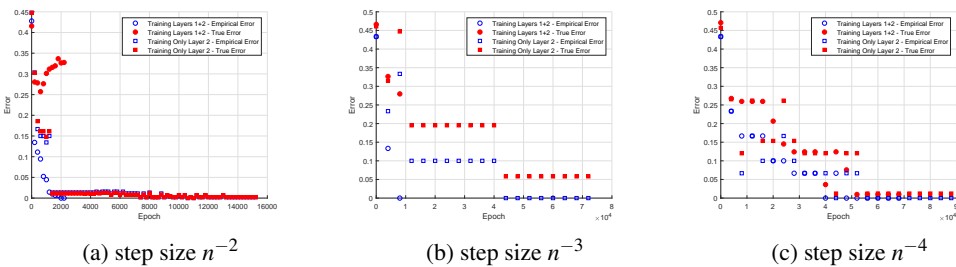

(a) step size $n^{-2}$
(b) step size $n^{-3}$
(c) step size $n^{-4}$

Figure 3: Error during training for an input dimension and a sample of size $n = 30$.

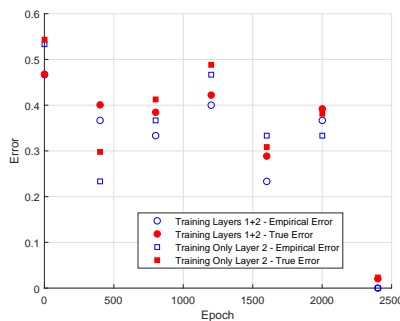

Figure 4: $\beta = 0$: error during training for an input dimension and a sample of size $n = 30$.

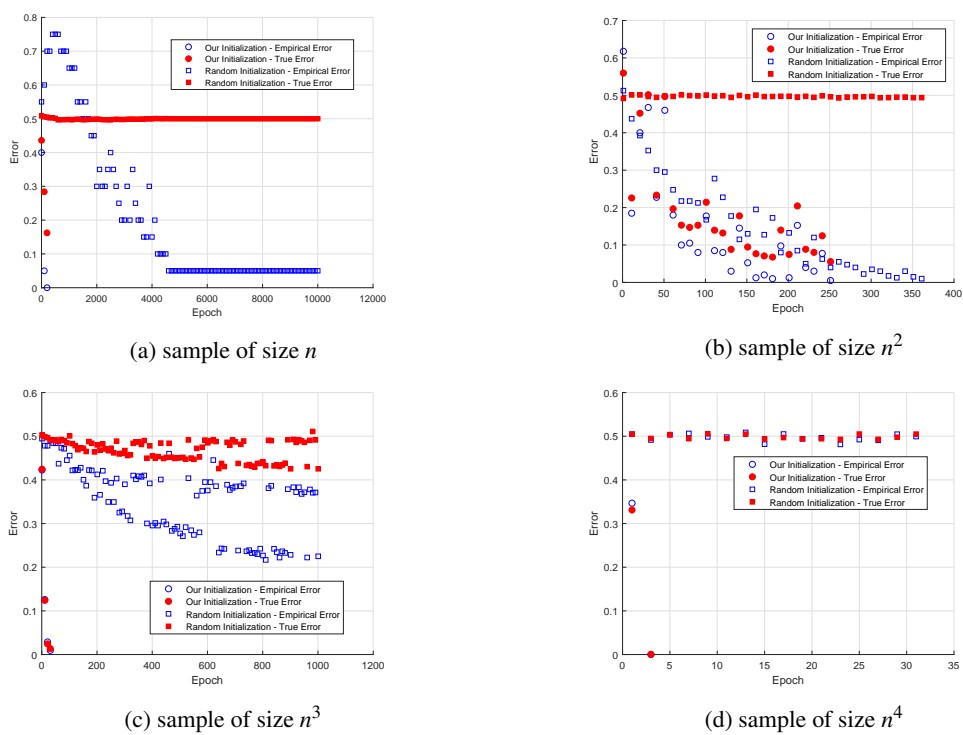

(a) sample of size $n$

(b) sample of size $n^2$

(c) sample of size $n^3$

(d) sample of size $n^4$

Figure 5: Parity: error during training for input dimension $n = 20$.

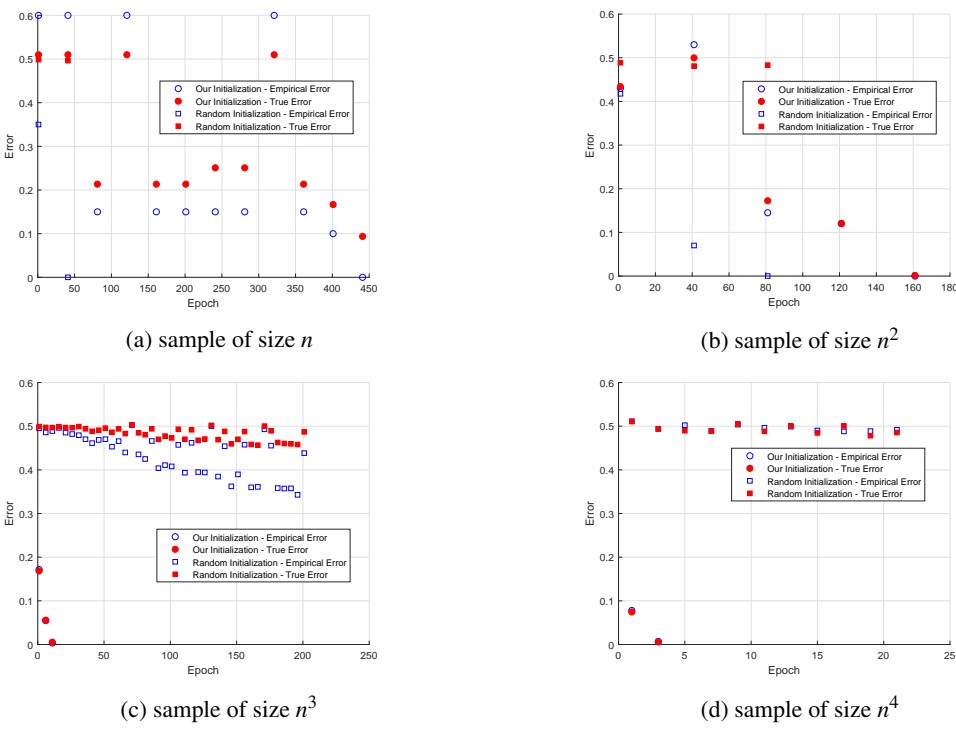

(a) sample of size $n$

(b) sample of size $n^2$

(c) sample of size $n^3$

(d) sample of size $n^4$

Figure 6: Random symmetric function: error during training for input dimension $n = 20$.

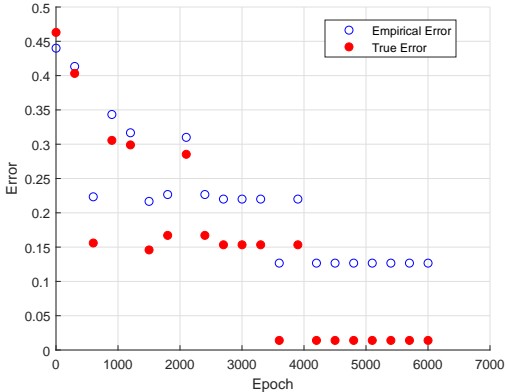

Figure 7: Label error resistance. Labels of the sample were flipped with probability $p = \frac{1}{10}$. Sample of size $10n$ whose input dimension is $n = 30$.

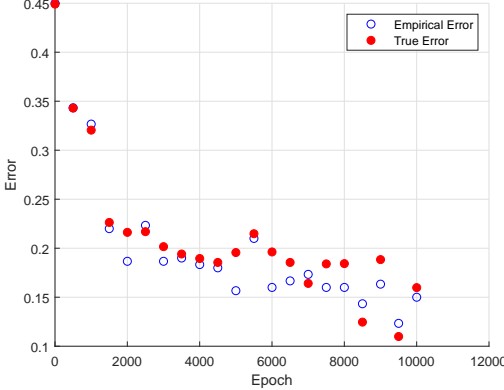

Figure 8: Input Error resistance. All the entries of the vectors in the space were randomly shifted. Sample of size $10n$ whose input dimension is $n = 30$.

