# OpenReview forum: "On Symmetry and Initialization for Neural Networks"
_ICLR.cc/2020/Conference — Reject_

### Official Review · AnonReviewer2 · 2019-10-22
**Official Blind Review #2**

**Rating:** 3

**Review:**

The paper studies the problem of learning the class of symmetric boolean function, that is, functions that depend only on |x| = \sum_i x_i. The paper shows that with proper initialization, one-hidden layer over-parametrized networks can learn this class of functions. The main observation that the authors make is that the last layer weights are updated as in the Perceptron algorithm and as long as the first layer has learned a large-margin representation, the first-layer weights do not change much. The authors experimentally validate their theory and additionally show that random initialization fails to converge to a low test-error solution while their special initialization works.

Overall, the main complexity arises from handling training of both layers and this is cleverly analyzed. However I am leaning towards rejection as the underlying problem does not seem well-motivated. The class of symmetric boolean functions can be modeled as a univariate function by mapping x -> |x| which is easy to solve in the no noise setting analyzed by the paper. Also, in terms of learning with neural networks, as the authors point out, one can learn this class by training only the last layer. It is unclear why this setup warrants the use of a neural network for training. The problem would be more challenging and interesting for the class of symmetric (permutation invariant) functions on the real domain where using symmetry in the architecture/initialization can potentially give gains.

Writing - Proofs are mostly clear however it would help to add more details in the proof of the main theorem (especially to argue about the use of the Perceptron convergence theorem for the changing representations). Also, the introduction needs to further motivate the setup and its relevance to neural networks.

Representation - Regarding the representation for indicators using ReLUs, one could use a simpler and more standard representation. In prior work the indicator is represented using a difference of ReLUs, 1[|x| >= i] = ReLU(|x| - i + 1) - ReLU(|x| - i) and 1[|x| = i] = 1[|x| >= i] - 1[|x| >= i+1] = ReLU(|x| - i + 1) - 2 ReLU(|x| - i) + ReLU(|x| - i - 1). Now one can express \sum_{i \in A} 1[|x| = i] by summing up these indicators and adding a bias term of -0.5 will make the sign be the correct value. Note that this would overall require only n + 2 hidden units with the weights now being bounded by constants. This would still have a margin of \Omega(1/n). Is there a particular reason for the choice of representation in the paper?

Experiments - The plots are hard to parse and inconsistent. Firstly, it would be better to use line plots instead of scatter plots to highlight the trend. Secondly, the x-axis needs to be sampled more frequently. The number of epochs seem to be varying in the experiments, please make that consistent. Lastly, the important plots need to be moved to the main paper. Are the experiments for multiple runs or just a single run?

**Experience Assessment:**

I have published in this field for several years.

**Review Assessment: Checking Correctness Of Derivations And Theory:**

I assessed the sensibility of the derivations and theory.

**Review Assessment: Checking Correctness Of Experiments:**

I assessed the sensibility of the experiments.

**Review Assessment: Thoroughness In Paper Reading:**

I read the paper at least twice and used my best judgement in assessing the paper.

---

> ### Author Response · Authors · 2019-11-07
> **Response to Review #2**
>
> Thank you for your time reading our work and for your feedback. Below, we try to clarify your concerns and we’ll be happy to hear your perspective again after you read our response.
>
> Your main issue is similar to Review #3, so please also refer to our comments there.
>
> “It is unclear why this setup warrants the use of a neural network for training.”
>
> It does not warrant it. The class is easily learnable with no special algorithm. Once you receive an input of a given weight you immediately know the label of all vectors with the same weight. It is just a matter of collecting enough vectors with different weights.
>
> We study this problem to have a fine-grained analysis of neural networks for a specific class of functions. Such analysis does not appear in the literature, as far as we know. Notice that this theoretical result demonstrates that although the neural network is over parametrized, by using SGD it still generalizes well.
>
> From an empirical and theoretical perspective, it is surprising that such a simple class of functions is not learnable by standard neural networks.
>
> “The class of symmetric boolean functions can be modeled as a univariate function by mapping x -> |x| which is easy to solve in the no noise setting analyzed by the paper.”
>
> Potentially, this mapping will not work in the noisy case. We show empirically that our initialization works in two different noisy cases.
>
> “Writing - Proofs are mostly clear however it would help to add more details in the proof of the main theorem (especially to argue about the use of the Perceptron convergence theorem for the changing representations). “
>
> Are you maybe referring to the same issue about Lemma 4 Review#3 had? Maybe our response can clarify things. Indeed, Lemma 4 has a bit of a tricky proof.
> If not, can you specifically refer us to arguments/sentences that were not clear?
>
>  “Regarding the representation for indicators using ReLUs, one could use a simpler and more standard representation…”
>
> This is indeed a simpler representation and we can use it. We didn’t use it because we simply were not aware of it. We just used the representation that we naturally found ourselves. Thank you for introducing it to us.
>
> “The number of epochs seem to be varying in the experiments, please make that consistent.”
>
> The focus of this work is the number of SGD updates. So if the number of samples is different for every case (n, n^2, n^3, n^4), it is only natural to have fewer epochs for larger datasets.
>
> “Lastly, the important plots need to be moved to the main paper.”
>
> We agree. They were put there because of the page limitation.
>
> “Are the experiments for multiple runs or just a single run?”
>
> A single run, that represents all other runs we experimented with while working in this setting. We observed the same behavior on all of our runs.

---

> > ### Comment · AnonReviewer2 · 2019-11-14
> > **Response**
> >
> > Thanks for addressing my comments. I did have the same issue as the other reviewer. Would be useful to make it explicit in the proof. As for the overall class of functions the paper is trying to learn, I'm still not convinced it gives us any new insight on how neural network training works since post initialization it is equivalent to learning a linear classifier. It would help the paper to extend the techniques developed to other more complex classes.

---

### Official Review · AnonReviewer3 · 2019-10-29
**Official Blind Review #3**

**Rating:** 3

**Review:**

PAPER SUMMARY: This paper studies the problem of training a single hidden layer neural network to represent an arbitrary symmetric function. These are functions $f : \{0,1\}^n \to \{-1, 1\}$ which are invariant to permutations in the input coordinates. The authors' main result (Theorem 1) shows that if you take a single hidden layer network with $O(n)$ hidden units and initialize the weights in a particular way, then for any symmetric $f$, SGD training will converge to an empirical risk minimizer with guaranteed small generalization error. On the other hand, the authors' experiments suggest that arbitrary symmetric functions are not learnable from random initialization. Taken together, these results point to the importance of designing network architectures/ initializations that respect the structure in the function class you're trying to represent.

REVIEW SUMMARY: I lean towards rejecting this paper however, because I am not convinced of the results' significance. We already know how to learn symmetric functions (see Exercise 3.26 in Mohri et al., 2018). The authors' results show that we can inject this knowledge into a neural network at initialization, and then run SGD without making things too much worse. I do not see how these ideas might apply to more substantial learning problems where our prior knowledge is less precise. Moreover, while the proofs are clearly presented overall, I have one concern with a key step in Lemma 4.

MAJOR COMMENTS:

1) The key property of symmetric functions is that their output depends only on $|x|$. Thus, one can first extract "cardinality features" $x \mapsto |x|$, after which learnability follows by standard generalization theory results (as the authors note in the proof of Theorem 1).

The basic idea of Theorem 1 then seems to be to realize this feature map as the hidden layer of a single hidden layer ReLU network (this is essentially what the initialization does) and then show that running SGD will not move the weights too far from the initialization (Lemma 4).

(a) First, I think it would be helpful to the reader if the authors could make this intuition more explicit. In the submission the authors do not give much explanation for the choice of initialization.

(b) Second, because this is a learning problem we already know how to solve, the results seems a little contrived. I do not see how these ideas could extend to more challenging cases where our prior knowledge of symmetry (e.g. translation invariance) does not by itself lead to an algorithm with efficient learnability guarantees.

2) I could not follow one step in the proof of Lemma 4 (used to show that SGD does not move the weights too far from the initialization). Why does Theorem 2 imply that the number of updates is at most $20 R^2 / \gamma^2$? In Theorem 2, $R$ is fixed whereas in Lemma 4 it varies with $t$. To me this seems important, since without a bound on the number of steps it is unclear how you can control how far the embeddings move.

MINOR COMMENTS

3) In the statement of Lemma 4, linear separability of $V$ should be with respect to some fixed partition $Y$?.

4) In Figure 5, why is empirical error not decreasing over epochs?

5) I think the figures referenced in the text should be in the paper, not the appendix.

Mohri, M., Rostamizadeh, A., & Talwalkar, A. (2018). Foundations of machine learning. MIT press.

**Experience Assessment:**

I have read many papers in this area.

**Review Assessment: Checking Correctness Of Derivations And Theory:**

I carefully checked the derivations and theory.

**Review Assessment: Checking Correctness Of Experiments:**

I carefully checked the experiments.

**Review Assessment: Thoroughness In Paper Reading:**

I read the paper at least twice and used my best judgement in assessing the paper.

---

> ### Author Response · Authors · 2019-11-07
> **Response to Review #3**
>
> Thank you for your time reading our work and for your feedback. Below, we try to clarify your concerns and we’ll be happy to hear your perspective again after you read our response.
>
> Regarding the results’ significance, we are not aware of any previous result that proves that a class of functions is PAC learnable using neural networks and SGD and considers all real-life elements of the training and generalization. For example, training all layers simultaneously, working with a fixed dataset, considering generalization and not just the training error, etc. We state the differences between our work and others’ in the Related Work section.
>
> As a fine-grained analysis of neural networks (such as appears in the paper) should start somewhere, working with the simple class of symmetric functions is maybe a good start. It wasn’t the point to learn symmetric functions but to understand better the dynamics of training neural networks. Our analysis of SGD shows that it has an optimal sample complexity (in terms of the VC-dimension of the class of symmetric functions) and that SGD is also efficient in time and memory.
>
> Also, Theorem 1 demonstrates that although the network has much more parameters \Omega(n^2) than samples O(n), it is still able to generalize well (even when all weights are allowed to change). A phenomenon that happens in practice and not well explained.
>
> We also find the empirical evidence surprising that a neural network cannot learn a random symmetric function from a random initialization, although these functions have a lot of structure and are easy to learn. This suggests another line of research; why these simple functions are hard to learn?
>
> Finally, Lemma 4 holds in general. It is not specific for symmetric functions and our initialization. It can be of theoretical interest independently of the paper.
>
> From a practical perspective:
> 1.	Empirically, we show that our initialization is robust to different types of noise. This suggests that our initialization can maybe help in cases where the function we learn is “highly symmetric” but not entirely.
> 2.	Lemma 4 suggests that if during training the neural network found a good embedding of the original space and the weights of the output neuron are not too large, then running SGD will converge to a network with a small empirical error that also generalizes well. This suggests a practical idea to regularize only the weights of the output neuron. Doing this will make sure the network will not “take a pass” on a good embedding of the original space.
>
> We didn’t add these points, as it sidetracks the theoretical analysis and the purpose of the paper.
>
> Regarding the other comments:
>
> “I could not follow one step in the proof of Lemma 4 (used to show that SGD does not move the weights too far from the initialization). Why does Theorem 2 imply that the number of updates is at most 20R^2/gamma^2? In Theorem 2, is fixed whereas in Lemma 4 it varies with. To me this seems important, since without a bound on the number of steps it is unclear how you can control how far the embeddings move. “
>
> This is indeed important and is taken into consideration in the paper. Let us clarify: We first prove that if at most $20R^2/gamma^2$ updates were made, then the norm of all $v$ is at most $2R$ while keeping a margin of 0.9\gamma. Now, we can use Theorem 2, which bounds the number of possible updates only in terms of the margin (0.9\gamma) and maximal norm (2R) (independent of the size of the dataset).
>
> “In the statement of Lemma 4, linear separability of V should be with respect to some fixed partition Y?”
>
> Yes. X is partitioned and this induces a partition of V. We assume that at the beginning, V is linearly separable and under the right conditions, it stays this way (although V changes).
>
> “First, I think it would be helpful to the reader if the authors could make this intuition more explicit. In the submission the authors do not give much explanation for the choice of initialization. “
>
> Yes, the purpose of the initialization is to embed the original space of binary vectors in such a way that every symmetric function linearly partitions the embedded space.
>
> “In Figure 5, why is empirical error not decreasing over epochs? “
>
> The empirical error does not decrease because we have more samples (n^4) than parameters (O(n^2)). The optimization problem is now hard. The bigger the sample, the harder it is for the network to fit the data. You can observe this gradual increase in hardness the bigger the sample you take.
>
> "I think the figures referenced in the text should be in the paper, not the appendix."
>
> We agree. They were put there because of the page restriction.

---

### Author Response · Authors · 2019-11-15
**A final comment to all reviewers**

We uploaded a revised version of the paper.
1. We modified the representations of the symmetric functions to the simpler representations suggested by Reviewer#2.
2. We modified some phrasing in the proof of Theorem 1, to make it clearer.

Recently, we came to know this work: https://arxiv.org/abs/1910.06956 and some several works that followed it.
It studies the behavior of neural networks when dealing with an infinite width network while only considering the training with continuous GD and its empirical error. In essence, these works study training of neural networks where the embedding of the original space by the network does not change with time much. These works attracted considerable attention. So it seems we don't have enough understanding of the dynamics in these regimes yet.

Our work is an example of the above phenomenon and studied in full detail: finite network, discrete SGD on all layers, polytime, and generalization guarantees.  Additionally, Lemma 4 gives the tools to study the training process of a finite width neural network performing discrete SGD (not only the special initialization we suggested). Also, Lemma 4 can enable to derive some generalization guarantees.

---

### Decision · Program_Chairs · 2019-12-19

**Decision:**

Reject

**Comment:**

The two main concerns raised by reviewers is that whether the results are significant, and a potential issue in the proof. While the rebuttal clarified some steps in the proof, the main concerns about the significance remain. The authors are encouraged to make this significance more clear.

Note that one reviewer argued theoretical papers are not suitable for ICLR. This is false, as a theoretical understanding of neural networks remains a key research area that is of wide interest to the community. Consequently, this review was not considered in the final evaluation.